# Cannabinoids in Breast Cancer: Differential Susceptibility According to Subtype

**DOI:** 10.3390/molecules27010156

**Published:** 2021-12-28

**Authors:** Cristina Ferreira Almeida, Natércia Teixeira, Georgina Correia-da-Silva, Cristina Amaral

**Affiliations:** 1Laboratory of Biochemistry, Department of Biological Sciences, Faculty of Pharmacy, University of Porto, Rua Jorge Viterbo Ferreira, n° 228, 4050-313 Porto, Portugal; up201404099@fc.up.pt (C.F.A.); natercia@ff.up.pt (N.T.); 2Associate Laboratory i4HB—Institute for Health and Bioeconomy, Faculty of Pharmacy, University of Porto, Rua Jorge Viterbo Ferreira, n° 228, 4050-313 Porto, Portugal

**Keywords:** cannabinoids, *Cannabis sativa*, anandamide, Δ^9^-tetrahydrocannabinol, cannabidiol, cannabigerol, cannabinol, cannabidivarin, breast cancer

## Abstract

Although cannabinoids have been used for centuries for diverse pathological conditions, recently, their clinical interest and application have emerged due to their diverse pharmacological properties. Indeed, it is well established that cannabinoids exert important actions on multiple sclerosis, epilepsy and pain relief. Regarding cancer, cannabinoids were first introduced to manage chemotherapy-related side effects, though several studies demonstrated that they could modulate the proliferation and death of different cancer cells, as well as angiogenesis, making them attractive agents for cancer treatment. In relation to breast cancer, it has been suggested that estrogen receptor-negative (ER^−^) cells are more sensitive to cannabinoids than estrogen receptor-positive (ER^+^) cells. In fact, most of the studies regarding their effects on breast tumors have been conducted on triple-negative breast cancer (TNBC). Nonetheless, the number of studies on human epidermal growth factor receptor 2-positive (HER2^+^) and ER^+^ breast tumors has been rising in recent years. However, besides the optimistic results obtained thus far, there is still a long way to go to fully understand the role of these molecules. This review intends to help clarify the clinical potential of cannabinoids for each breast cancer subtype.

## 1. Cannabinoids: Overview

*Cannabis sativa*, a member of the Cannabaceae family commonly known as marijuana and originated from Central Asia, has been used for more than 5000 years, being one of the world’s oldest plant sources of medicines and textile fiber [1,2,3]. Its therapeutic actions were first described in India, due to its analgesic, antiemetic and anticonvulsant properties [4]. Currently, *Cannabis sativa* is also widely used for recreational purposes, being considered the number one illicit drug in several countries [5,6]. Moreover, in recent years, this plant has gained a significant clinical interest and its medical use is already regulated by law in several countries [7].

Until now, more than 500 compounds from *Cannabis sativa* have been characterized, including cannabinoids, flavonoids, terpenes and fatty acids, present in the leaves and buds of the plant [1,3,8,9]. About 100 of those compounds, accounting for about 24% of all products of the plant, are phytocannabinoids (Figure 1), Δ^9^-tetrahydrocannabinol (THC) being the main psychoactive cannabinoid [3,10,11]. It is important to note that the term cannabinoids refers not only to phytocannabinoids, but also to endogenous cannabinoids, the endocannabinoids (Figure 1), such as N-arachidonoylethanolaminee (anandamide, AEA) and 2-arachidonoylglycerol (2-AG), and to synthetic cannabinoids [3]. The latter are a heterogeneous group of compounds, that were developed to be used in research studies and as potential therapeutic agents. They are classified into several structural groups, including adamantoylindoles, aminoalkylindoles, benzoylindoles, cyclohexylphenols, dibenzopyrans, naphtoylindoles, naphthylmetylindoles, naphthylmethylindenes, naphthoylpyrroles, phenylacetylindoles, tetramethylcyclopropyl ketone indoles, quinolinyl ester indoles and indazole carboxamide compounds. Furthermore, they have higher affinity for CBs than phytocannabinoids and endocannabinoids, acting, usually, as direct agonists. Because of their high potency, they are associated with more adverse effects than the natural cannabinoids. Some of those compounds such as JWH-133, Met-F-AEA, JWH-015 and WIN 55, 212-2 have already been evaluated in some pathologies, including breast, lung, colon, liver and prostate cancers, where they showed promising antitumor effects [12,13,14].

Besides THC, cannabidiol (CBD) is another well-known phytocannabinoid that has gained a lot of attention in recent years because of its therapeutic potential. This is, in part, due to the absence of psychotropic activity. In the plant, THC and CBD are present as acidic precursors, Δ^9^-tetrahydrocannabinolic acid (THCA) and cannabidiolic acid (CBDA), respectively, being converted to THC and CBD, through decarboxylation induced by increased temperatures [2]. In addition to these two main phytocannabinoids, there are also other phytocannabinoids, designated as minor phytocannabinoids, that, despite being less studied, may present interesting pharmacological effects. Some of those compounds are cannabinol (CBN), the first cannabinoid isolated, cannabigerol (CBG) and cannabidivarin (CBDV) [2,10].

The plant *Cannabis* can present three different phenotypes: phenotype I or drug type, phenotype II or intermediate phenotype and phenotype III or fiber type or hemp. Empathizing the importance and abundance of THC and CBD, phenotype I is characterized by a higher proportion of THC, while intermediate phenotype II is known to contain a higher concentration of CBD and variable concentrations of THC, and phenotype III presents CBD as the primary phytocannabinoid [15].

Thus, it is strongly believed that *Cannabis*, being an important natural source of many cannabinoids, may be a potential therapeutic option for the treatment or modulation of different physiological processes and, even, pathological conditions, such as cancer.

## 2. Cannabinoid Receptors

Contrary to what was initially thought, cannabinoids exhibit a high stereo selectivity [16,17,18], interacting with specific receptors designated as cannabinoid receptors (CBs). The existence of these receptors was demonstrated in 1984 by Allyn Howlett [19], when she showed that THC was able to decrease cAMP levels in cell cultures of neuroblastoma. In fact, CBs belong to the G protein-coupled receptor family, trigger mitogen-activated protein kinases (MAPKs) and inhibit adenylyl cyclase activity, impairing protein kinase A (PKA) activation [3]. Two CBs, CB1 and CB2, which share amino acid sequence homology of 44% [20], have been identified [21,22]. CB1, encoded by the *CNR1* gene, which maps to chromosome 6q14–q15 [23,24], was isolated for the first time in 1990 from the rat brain [25] and is mainly expressed in the central nervous system (CNS) [21,26]. Additionally, it is also expressed in some peripheral tissues, including the gastrointestinal tract, liver, adipose tissue and cardiovascular and reproductive systems [3,26]. At the cellular level, CB1 is mainly present on the plasma membrane, but it can also be found in the membranes of endosomes, lysosomes and mitochondria. Considering its expression pattern, CB1 is involved in several functions, such as mood, memory, sensation, cognition, motor coordination and autonomic function [21]. Besides these functions, CB1 also displays key roles in the central and peripheral regulation of food intake, in fat accumulation and in the metabolism of lipids and glucose [21,27]. Regarding cancer cases, CB1 expression is known to be increased in prostate, pancreatic and colon cancers, hepatocellular carcinoma, non-Hodgkin lymphoma and astrocytoma [28].

CB2 is encoded by the *CNR2* gene located on chromosome 1p36 and was first cloned in 1993 from human promyelocytic leukemia HL-60 cells [22]. This cannabinoid receptor is essentially expressed in peripheral cells and in the immune system, but it can also be found in the CNS, in microglia and neurons [2,3,21,29]. Regarding the immune system, CB2 is mainly expressed on leucocytes and cells from the spleen and tonsils, being able to modulate the migration of immune cells and release of cytokines [3,5,30]. The expression of CB2 is increased in breast cancer, hepatocellular carcinoma, glioma and astrocytoma [28].

The expression of CBs has already been described in several species, including human, monkey, pig, dog, rat and mouse [21], and the crystallographic structures have been recently resolved, the CB1 structure in 2016 [31] and the CB2 structure in 2019 [32]. This represents an important achievement regarding CBs function, since it may increase the knowledge related to their modulation.

Cannabinoids may also induce their actions through interaction with other receptors [5,21]. One alternative receptor is the orphan G protein-coupled receptor 55 (GPR55), which, in humans, is mainly expressed in the brain and liver [33]. GPR55 shares limited homology with CB1 (13%) and CB2 (14%), which is the reason why it was suggested as the third CB, CB3. However, there are still inconsistencies regarding its activation and modulation, representing a reason why more research is needed to classify it as a CB [5].

## 3. Endocannabinoid System

Since CBs were discovered, it has been speculated that endogenous ligands might exist. In fact, in 1992, the endocannabinoid AEA was isolated from the pig brain [34], and three years later, 2-AG was found in the canine gut [35] and rat brain (Figure 1) [36]. AEA is synthetized on demand from membrane phospholipids through the action of the enzyme *N*-acylphosphatidylethanolamine-phospholipase D (NAPE-PLD) [5,21,37]. This endocannabinoid exerts several physiological functions in the central and autonomic nervous systems, in the immune and reproductive systems, in the endocrine network and in the gastrointestinal tract [5,38]. It is responsible for analgesia, control of motor activity, reduction in emesis, appetite stimulation, hypothermia induction and antiproliferative effects [5]. AEA binds to CB1 and CB2 [5,39], presenting a lower affinity for the latter (approximately four-fold less than for CB1) [40,41], acting as a partial agonist in both receptors [5]. Besides CBs, AEA is also able to interact with other molecular targets, including the transient receptor potential vanilloid 1 (TRPV1) (Figure 2) [42] and the peroxisome proliferator-activated receptors (PPARs) [43]. AEA is very unstable, and, thus, it is rapidly hydrolyzed by fatty acid amide hydrolase (FAAH) into arachidonic acid and ethanolamine [10,44].

Similar to AEA, 2-AG is produced on demand from membrane phospholipids by the diacylglycerol lipase (DAGL) [45,46]. Its biological activities include modulation of the immune system, cell proliferation, embryo development, hippocampus long-term potentiation, neuroprotection, neuromodulation, cardiovascular function and inflammatory responses [47]. 2-AG presents higher affinity for both CB1 and CB2 than AEA, being considered the main endocannabinoid in the brain to act as a full agonist at CB1 [5,48]. Furthermore, 2-AG is also present at higher concentrations than AEA in the reproductive system [5]. Additionally, similar to AEA, 2-AG can interact with PPARs [49,50], but not with TRPV1 (Figure 2), being degraded by monoacylglycerol lipase (MAGL) and by the hydrolyzing enzymes alpha beta hydrolase domain-6 (ABHD6) and -12 (ABHD12) into arachidonic acid and glycerol [10,51]. It is important to note that the metabolization of both AEA and 2-AG can also occur through cyclooxygenase-2 (COX-2) and lipoxygenases (LOXs) [5].

In addition, other putative endocannabinoids have been identified, including 2-arachidonoylglyceryl ether (2-AGE) [52], virodhamine (*O*-arachidonoylethanolamine) [53], *N*-arachidonoyldopamine (NADA) [54], *N*-arachidonoyl glycine (NAGly) [55] and oleamide (ODA) [56]. However, their biosynthesis, as well as their ability to activate CBs, still needs to be fully elucidated [5].

The endocannabinoids together with the CBs, the metabolic enzymes responsible for endocannabinoid synthesis and degradation, and the endocannabinoid membrane transporter (EMT) constitute the endocannabinoid system (ECS) [5]. In recent years, several roles in the modulation of human physiological processes, including appetite, circadian rhythm, inflammation, stress, pain and reproduction, as well as pathophysiological situations, such as cancer, have been attributed to this system, making it a potential important target for the management of many conditions [6,57,58,59]. In fact, since this system is tightly regulated, any disturbance can lead to its deregulation, which can be beneficial or not, depending on the target disease/condition.

## 4. Phytocannabinoids

### 4.1. Δ^9^-Tetrahydrocannabinol

Δ^9^-tetrahydrocannabinol (THC) was first extracted from *Cannabis sativa* in 1942 by Wollner et al. [2], and its structure was elucidated only in 1964 by Yechiel Gaoni and Raphael Mechoulam [60]. Similar to AEA and 2-AG, this phytocannabinoid is able to activate CB1 and CB2 receptors, acting as a partial agonist, which may explain its psychotropic activities induced through the CB1 receptor [61,62]. Additionally, it is known that the activation of CB1 by THC is also associated with hypolocomotion, hypothermia, catalepsy and analgesia, the known “tetrad model” [63], while the activation of CB2 and PPARs is frequently linked to neuroprotective, antispasmodic and anti-inflammatory effects [64,65,66]. It has also been reported that THC can modulate other receptors besides CBs, such as transient receptor potential cation channels (TRP), GPR18 and GPR55 (Figure 2) [3]. THC is also responsible for the addictive properties of *Cannabis sativa*, since increased abuse and dependence have been related to its concentration [61,67].

### 4.2. Cannabidiol

Cannabidiol (CBD) was isolated in 1940 [68], and its structure was elucidated by Raphael Mechoulam in 1963 [69]. CBD presents low affinity for the CBs and acts as an inverse agonist, which may explain its lack of psychotropic activity and its ability to antagonize some THC-induced effects, such as anxiety, hunger, sedation and tachycardia [10,70]. Additionally, this phytocannabinoid can also act as an allosteric modulator of the CB receptors [71,72] or bind to other receptors, including GPR55 (Figure 2) [10,61]. Moreover, CBD is also able to interact with enzymes, such as those of the cytochrome P450 family (CYP450) and FAAH, the enzyme that hydrolyzes AEA [10]. However, the response of CBD on all these targets is dependent on its concentration and on the cell models [10]. Overall, different studies have attributed several effects to CBD, including anticonvulsive, neuroprotective, antioxidative, anti-inflammatory, analgesic and antiemetic effects [73,74].

### 4.3. Minor Phytocannabinoids

Cannabinol (CBN), a psychotropic cannabinoid initially thought to be the active psychotropic agent of cannabis [75], was isolated from cannabis at the end of the 19th century, and its structure was elucidated by Robert Cahn in the 1930s [2]. It has been suggested that THC is the precursor of CBN, the latter being produced during the storage of harvested cannabis [2]. This cannabinoid binds to CB1 and CB2, but with less potency than THC, displaying higher affinity for CB2 than for CB1 (Figure 2). Additionally, it has also been demonstrated that CBN can inhibit the enzymes CYPA1, CYP1A2 and CYP1B1, desensitize TRPA1 cation channels and block TRPM8 cation channels [3].

Cannabigerol (CBG) was first isolated in 1964 [76], and its structure was proposed in 1971 by Gaoni and Mechoulam [77]. Little is known about this cannabinoid, but similar to CBD, CBG lacks the ability to induce psychotropic effects. It has been demonstrated that CBG can avoid the activation of CB1 and modulate the activity of other receptors (Figure 2) [3].

Cannabidivarin (CBDV) was isolated in 1969 and is a propyl analog of CBD [78]. Little is still known about the mechanism of action of this phytocannabinoid, but the lack of psychotropic activity and its anticonvulsive properties have already been demonstrated, making CBDV a promising therapeutic agent [79]. Similar to CBD, CBDV presents very low affinity for the CBs [80,81]. Furthermore, CBDV can act as an agonist of the TRPA1, TRPV1 and TRPV2 channels [82] and modulate the activity of other receptors, such as GPR55 (Figure 2) [83,84]. It was also shown that this cannabinoid inhibits the enzyme responsible for 2-AG biosynthesis, DAGL [82].

## 5. Cannabinoids and Breast Cancer

Cannabinoids have been applied for the treatment and management of several diseases and conditions, including pain, cramps, migraine, limb muscle spasms, asthma, sleep disorders, depression, insomnia and emesis [1,10,85]. During the last 20 years, intense research has been conducted in order to evaluate the clinical and pharmacological potential of cannabinoids, alone or in combination, for the treatment of different pathological conditions. In fact, some cannabinoid-based medicines are already approved for clinical use in many countries. Those treatments include nabiximols (Sativex^®^), a 1:1 mixture of THC and CBD used in multiple sclerosis [86], dronabinol (Marinol^®^) and nabilone (Cesamet^®^), two THC synthetic analogs used to relieve chemotherapy-related side effects, such as vomiting and nausea [7,87], and CBD oil (Epidiolex^®^) for the treatment of some pediatric epilepsy conditions, which was recently approved by the FDA [88]. Moreover, cannabinoids have been evaluated for other conditions and diseases, including cancer. In fact, considering the expression of endocannabinoids in vertebrate and invertebrate organisms and their ability to modulate the activity of proteins responsible for cell proliferation, differentiation and apoptosis, it was postulated that the ECS may be involved in the control of cell proliferation and survival [6]. However, there are conflicting data regarding the role in tumor development, since some studies suggest that it may be overactivated [89], while others demonstrate that the activation of CBs reduces tumor growth, suggesting that the ECS may display tumor-suppressive actions [59,90]. This biphasic effect seems to be dose dependent, as low doses of cannabinoids promoted cell proliferation, while higher doses induced antiproliferative actions [91]. The antiproliferative effects of phytocannabinoids were demonstrated for the first time in 1975, by Munson et al., who showed that THC and CBN inhibit lung adenocarcinoma cell growth [92]. Over the years, other cannabinoids, including the endocannabinoids AEA and 2-AG and the synthetic cannabinoids WIN 55212-2 and HU-210, have also been highlighted for their antitumor actions in vitro and in vivo [59].

The primary antitumor effects of cannabinoids rely on cell cycle arrest through the inhibition of the expression of growth factors and induction of apoptosis. Additionally, they can also avoid angiogenesis and block invasion and metastasis by impairing the activation of the vascular endothelial growth factor (VEGF) pathway. Until now, the actions of cannabinoids have been identified in different tumors, including gliomas, melanomas, lymphomas, breast cancer, skin cancer, lung carcinoma, liver cancer, pancreatic cancer, colon cancer and prostate cancer, indicating that the antitumor actions are not tumor type specific [6,59,91]. In fact, several clinical trials have already been conducted or are currently underway to evaluate the safety and effectiveness of cannabinoids or cannabinoid-based preparations alone (NCT02255292, NCT01489826, NCT01654497, NCT04001010, NCT03617692, NCT03948074, NCT00316563, NCT03564548, NCT02802540, NCT04988490, NCT03245658, NCT04808531, NCT03984214, NCT03661892, NCT02073474, NCT00314808, NCT03944447, NCT02054754 and NCT00530764) or combined with chemotherapy agents such as temozolomide, bortezomib, fluorouracil, oxaliplatin, leucovorin, bevacizumab, irinotecan, palonosetron and dexamethasone (NCT01812603, NCT01812616, NCT03529448, NCT03607643, NCT02423239, NCT04155008 and NCT00553059). The results obtained thus far indicate that, generally, cannabinoids are well-tolerated compounds, without significant side effects and with a high potential for the modulation of pain and chemotherapy-related side effects.

In relation to breast cancer, data from pre-clinical studies suggest that cannabinoids may be beneficial for the treatment of the best-known breast cancer subtypes. Taking into account the expression of specific molecular biomarkers, such as estrogen receptor (ER), progesterone receptor (PR) and human epidermal growth factor receptor 2 (HER2), breast cancer can be divided into four subtypes, luminal A, luminal B, human epidermal growth factor receptor 2 positive (HER2^+^) or triple negative (TNBC) (Table 1) [93,94,95,96,97]. Luminal tumors are the most common, representing 60–73% of all cases. Luminal A is a low-risk breast cancer subtype, being low grade and presenting the most favorable prognosis. This subtype has a high ER-regulated gene expression and a high PR expression and does not express HER2, being responsive to endocrine therapy. On the other hand, luminal B breast cancers are high grade and, thus, more aggressive and associated with a worse prognosis than luminal A tumors. HER2 expression is generally associated with this subtype, and the tumors are usually positive for the expression of ER and PR. In turn, HER2^+^ or non-luminal breast tumors represent 12–20% of all breast tumors. This subtype is typically high grade, associated with a high aggressiveness and poor prognosis, and is characterized by an overexpression of HER2 along with the lack of expression of ER and PR. Finally, TNBC accounts for approximately 15–20% of all breast cancers, is typically high grade, presents a high proliferation index and is associated with a high rate of local and distant recurrence and, consequently, with a poor prognosis, usually being the most aggressive. Moreover, it is characterized by the absence of ER, PR and HER2 expression [93,94,98,99,100,101,102]. Around 80% of all TNBCs are basal-like, which means that these tumors are enriched in cells expressing characteristic genes of normal basal cells [103].

Evidence obtained thus far has revealed that, as in other cancer types, the ECS is altered in breast cancer cases, being intimately associated with tumor aggressiveness. In fact, endocannabinoid concentrations and expression levels of CBs, and of the enzymes responsible endocannabinoid metabolism, are typically associated with cancer aggressiveness, reinforcing their involvement in cancer development [104]. It is known that CB2 is overexpressed in breast cancer and that CB1 is present in significant lower quantities. CB2 expression is mainly observed in HER2^+^ tumors, being detected in 90% of all HER2^+^ tumors [105]. In this case, overexpression of CB2 is linked to a poor prognosis. In fact, a correlation has been established between CB2 expression and tumor aggressiveness, as mRNA CB2 levels were higher in ER^−^/PR^−^ tumors than in ER^+^/PR^+^ tumors, as well as in HER2^+^ tumors than in HER2^−^ tumors, and in high-grade histological tumors than in low-grade histological tumors [10,91]. On the other hand, CB2 expression on ER^+^ and ER^−^ tumors is associated with a better prognosis. In relation to endocannabinoids, in breast cancer cases, an increase in the levels of N-acylphosphatidylethanolamine, the AEA precursor, and in MAGL levels has been observed, mainly in ductal carcinomas [106,107,108].

### 5.1. Cannabinoids in Triple-Negative Breast Cancer

The majority of the studies conducted on breast cancer were performed on TNBC models. In fact, it has been established that the sensitivity of human breast cancer cell cultures to cannabinoids is correlated with their aggressiveness, being ER^−^ cell lines more sensitive than ER^+^ cells [91].

CBD is the most studied phytocannabinoid in TNBC (Table 2). It has been reported that this phytocannabinoid reduces the proliferation of MDA-MB-231 cells through the direct activation of TRPV1 receptors and possibly through other yet uncharacterized CBD targets [109,110]. However, it was also proposed that CBD induces apoptosis in this cell model through the involvement of CB1, CB2 and TRPV1 receptors. This effect is mediated by endoplasmic reticulum stress and inhibition of the AKT/mTOR pathway, which ultimately culminates in autophagy and mitochondria-driven apoptosis (Figure 3A) [111]. In fact, this mechanism was already identified in other models, suggesting that this is a general mechanism of action [91]. The induction of apoptosis by CBD in MDA-MB-231 cells was also recently verified by Sultan et al., where CBD inhibited cell survival and induced apoptosis, favored by an interplay among PPAR, mTOR and cyclin D1 (Figure 3A) [112]. In contrast, regarding the MDA-MB-231 cell line, a study showed that THC promotes the proliferation of these cells [113]. Mohamad Elbaz et al. also demonstrated antiproliferative as well as antimigratory and anti-invasive properties of CBD through the inhibition of EGF/EGFR signaling in several TNBC cell lines, including SUM159, 4T1.2 and SPC2 cells. In this case, the effects were mediated by downstream inhibition of the Raf-1/MEK/ERK, NF-kB and AKT signaling pathways (Figure 3A) [114]. In addition, CBD effects on cell proliferation were also verified in xenografts generated from MDA-MB-231 cells in immune-deficient mice and in orthotopic xenografts generated from 4T1 cells in syngeneic BALB/c mice. In both cases, CBD reduced tumor growth, but for the latter, acquired resistance to CBD was developed [110,115]. In addition, CBD also impairs the metastatic potential of MDA-MB-231 and 4T1 cells, probably by downregulation of Id-1, which is mediated by ERK (Figure 3A). Regarding this, it is important to note that the effects exerted by ERK signaling depend on the duration of the stimulus, since sustained ERK activation causes cell growth inhibition, while short-term stimulation promotes cell growth [115,116]. Moreover, it has been demonstrated, in MCD-MB-231 cells, that CBD induces the production of reactive oxygen species (ROS) [110,111,115,116], which are able to downregulate Id-1 activity (Figure 3A), corroborating the apoptotic effects previously mentioned. Nevertheless, a dependence on increased ROS levels for the survival of TNBC cell models has been verified [117]. In fact, ROS are known to play a key role in the breast cancer microenvironment, promoting the differentiation of neighboring cells, including fibroblasts that secrete growth factors, cytokines and metalloproteinases, leading to tumor development and growth [118,119]. Interestingly, the dependence on ROS levels seems to be particularly high in TNBC [117]. Considering this, several antioxidant and mitochondria-targeted therapies have been suggested for breast cancer [119].

The minor phytocannabinoids CBN and CBG also showed interesting results in TNBC. In MCD-MB-231 and MDA-MB436 cells, both compounds reduced cell viability and cell migration through a process probably involving a decrease in Id-1 expression [116] (Table 2). Combinations of minor and major cannabinoids have also been evaluated in TNBC [120].

Besides the actions of the phytocannabinoids, synthetic cannabinoids have also demonstrated interesting effects on TNBC (Table 2). In two in vivo models, a xenograft-based and a PyMT genetically engineered model of TNBC, the synthetic cannabinoid JWH-133 provoked a significant reduction in tumor growth and inhibited angiogenesis [121]. In fact, antimigration effects were induced by several synthetic cannabinoids, Met-F-AEA, WIN 55,212-2, JWH-133 and JWH-015, in MDA-MB-231 cells [121,122,123,124]. These effects were mediated by CB1, with inhibition of the focal adhesion kinase (FAK)/Src and RhoA-ROCK pathways [122,123], and by CB2, along with COX-2/PGE_2_ axis inhibition [121], or through inhibition of the ERK pathway and cytoskeletal adhesion and stress fiber formation [124]. Met-F-AEA, WIN 55,212-2 and JWH-133 also induce cell cycle arrest, [121,123]. These studies suggest that the interaction of synthetic cannabinoids with CBs reduces the metastatic potential of TNBCs, a behavior already identified for the formation of lung metastasis [122].

Therefore, in vivo and in vitro studies have shown that cannabinoids exert antiproliferative and antimetastatic actions on TNBC models, mainly by the induction of apoptosis and autophagy via CB activation, as well as modulation of several signaling pathways involved in cell proliferation, such as AKT/mTOR and EGF/EGFR.

### 5.2. Cannabinoids in Human Epidermal Growth Factor Receptor 2-Positive Tumors

In relation to HER2^+^ breast cancer, overexpression of CB2 has been verified, which is associated with a poor prognosis [105]. Moreover, in HER2^+^ breast cancer models, the antitumor effects are mediated by CB2 activation, which is the reason why CB2-directed therapy should be effective in growth inhibition of these tumors [91]. Therefore, cannabinoids represent an attractive potential therapy for this subtype of breast cancer. Until now, the most prominent study regarding HER2^+^ breast cancer and cannabinoids was conducted in MMTV-neu mice, a good model for the study of HER2^+^ tumors. In this study, it was revealed that THC impaired tumor growth, angiogenesis and the formation of metastasis through the induction of apoptotic cell death and inhibition of AKT (Figure 4A), effects that were reproduced with JWH-133, a selective CB2 agonist [125].

### 5.3. Cannabinoids in Luminal A Tumors

Regarding luminal tumors, the most frequently diagnosed are ER^+^ tumors. Studies have been developed in order to explore the effects of cannabinoids, and some conclusions have already been reached (Table 3). A study involving MCF-7 and EFM-19 cells demonstrated that AEA inhibits basal-, prolactin- and nerve growth factor (NGF)-induced proliferation through CB1 activation. Cell cycle arrest and activation of the Raf-1/ERK/MAPK pathway through the inhibition of adenylyl cyclase have been verified, since the sustained activation of the ERK pathway leads to the downregulation of prolactin and NGF receptors [91,126] (Figure 5). In fact, Melck et al. also showed that AEA reduces cell proliferation via CB1 activation, in MCF-7 cells [127,128]. On the other hand, THC induced a similar effect on EVSA-T cells, though this effect was mediated by the CB2 receptor through CDK1 inhibition, and activation of the transcription factor JunD [129,130] (Figure 4B). However, there are studies on MCF-7 cells showing that THC is also able to promote cell proliferation, an effect that might involve Act1 and Erb2 [113,131,132,133]. Regarding CBD, it was demonstrated in the T-47D cell line that this phytocannabinoid inhibits cell survival and promotes apoptosis due to an interplay among PPAR, mTOR and cyclin D1 [112] (Figure 3B), probably through a mechanism similar to the one observed in TNBC cases. Recently, it was also verified that CBD is able to induce endoplasmic reticulum stress, which leads to cell death, in MCF-7 cells through the activation or TRPV1 and the consequent increase in Ca^2+^ and ROS levels (Figure 3B) [134]. In relation to minor phytocannabinoids, CBG was shown to decrease MCF-7 cell proliferation [110], while CBN stimulated proliferation in the same cell line [131,132] (Table 3).

A very recent in silico study demonstrated that THC analogs possess the ability to bind to ERβ and, probably, lead to its activation [136]. This may represent an important advantage for ER^+^ breast cancer treatment, since this receptor is associated with antitumor effects. In fact, it has already been verified that THC prevents estradiol-induced proliferation in MCF-7 cells, in an ERα-independent manner [137]. Recently, our group also showed, in MCF-7aro cells, that AEA, CBD and THC reduce cell viability and disrupt cell cycle progression. Moreover, AEA and THC caused apoptotic cell death, while CBD induced autophagy to promote apoptosis. In addition, AEA and CBD were able to inhibit the enzyme aromatase [135], data that are in line with Almada et al.’s studies, where it was demonstrated that these cannabinoids inhibit aromatase in human placental microsomes [138,139]. In MCF-7aro cells, these three cannabinoids reduced aromatase and ERα levels, while only AEA and CBD were able to upregulate ERβ levels [135] (Figure 3B, Figure 4B and Figure 5), data that reinforce the interplay between the endocannabinoid system and estrogen signaling. In fact, one of the current fields of study regarding ER^+^ breast cancer and cannabinoids is their combination with endocrine therapy. Recently, it was demonstrated that tamoxifen and other SERMs act as CB1 and CB2 modulators, and Blasco-Benito et al. showed that tamoxifen in combination with THC or a cannabis preparation caused additive antiproliferative responses in T47D cells [140]. It was also demonstrated that tamoxifen binds to CBs, acting as an inverse agonist [141]. Furthermore, it was shown that THC upregulates ERβ [142], a benefic effect for ER^+^ breast cancer treatment. However, and contrary to the findings of our group, it was also reported that THC promotes MCF-7 cell growth, and that this effect is potentiated when combined with aromatase inhibitors (AIs) [133].

Additionally, combinations of major and minor cannabinoids evaluated on MCF-7 cells revealed pro-apoptotic effects [120].

In summary, in ER^+^ breast cancer, cannabinoids also have the potential to inhibit cell growth and avoid metastatic development. These effects are typically associated with cell cycle arrest, apoptosis and autophagy, through the involvement of CB1 and CB2 and the modulation of survival pathways, such as the mTOR and Raf-1/ERK/MAPK pathways.

## 6. Conclusions

Cannabinoids have been used for centuries in several therapeutic applications. Regarding cancer, the use of cannabinoids has already been approved in several countries for the relief of chemotherapy-associated effects, but their clinical potential is greater than initially thought, and their clinical interest has been rising in recent years. Pre-clinical studies have demonstrated that cannabinoids exert important antitumor properties in the main breast cancer subtypes, particularly in TNBC, where different phytocannabinoids and synthetic cannabinoids have shown interesting therapeutic actions. Therefore, more studies must be conducted, mainly on HER2^+^, luminal A and luminal B breast tumors, in order to better understand the mechanism of action of these compounds, which would help to clarify the therapeutic potential in breast cancer subtypes, and even in other cancer types. However, regarding luminal A breast cancer, some works have indicated that cannabinoids modulate key targets responsible for the survival and development of this breast cancer subtype, which are targeted by the therapies currently under clinical use. Moreover, as mentioned, despite the fact that most of the studies have attributed antitumor actions to cannabinoids, some have demonstrated that they can also exert pro-tumor effects. This biphasic behavior is correlated with the cannabinoid concentration used, since lower concentrations seem to induce cell proliferation and survival, while higher doses are associated with cell death and inhibition of cell growth. Therefore, it is important to keep that in mind in order to develop better therapies able to exert the desired effects. Despite this, most of the mechanisms of action induced by the cannabinoids appear to be common among the different breast cancer subtypes. Regarding clinical trials, those developed thus far are focused on the safety and effectiveness of cannabinoids in several cancer types, and, currently, there are no clinical trials focused only on breast cancer, what would be an asset to better understand the potential of these molecules in this type of cancer.

Thus, it is possible to conclude that, despite the need for more studies focused on breast cancer, cannabinoids are promising therapeutic agents for the different breast cancer subtypes, being able to exert important actions on cell survival and metastasis.

## Figures and Tables

**Figure 1 molecules-27-00156-f001:**
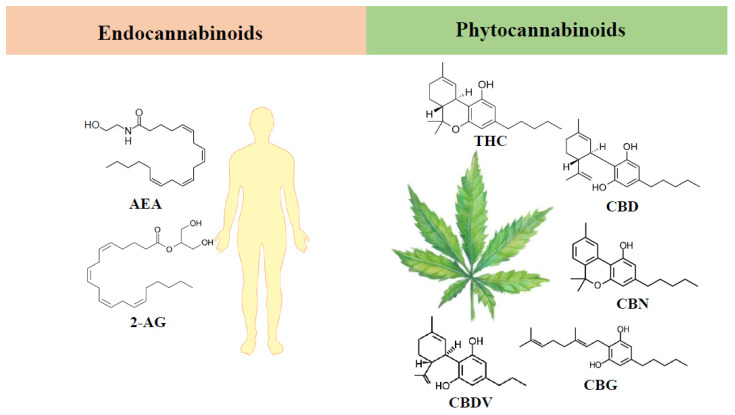
Chemical structures of the major endocannabinoids and phytocannabinoids. The endocannabinoids depicted are anandamide (AEA) and 2-arachidonoylglycerol (2-AG), and the phytocannabinoids are Δ^9^-tetrahydrocannabinol (THC), cannabidiol (CBD), cannabinol (CBN), cannabigerol (CBG) and cannabidivarin (CBDV).

**Figure 2 molecules-27-00156-f002:**
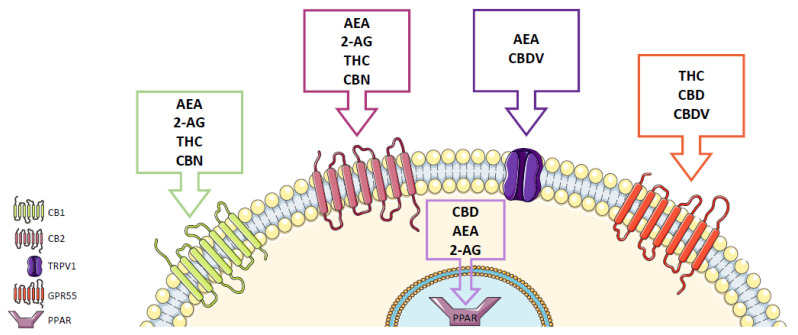
Cannabinoids and their receptors. The cannabinoid receptors CB1 and CB2, GPR55, as well as the cation channel TRPV1 and the nuclear receptor PPAR, are presented. AEA—anandamide; 2-AG—2-arachidonoylglycerol; THC—Δ^9^-tetrahydrocannabinol; CBD—cannabidiol; CBN—cannabinol; CBDV—cannabidivarin.

**Figure 3 molecules-27-00156-f003:**
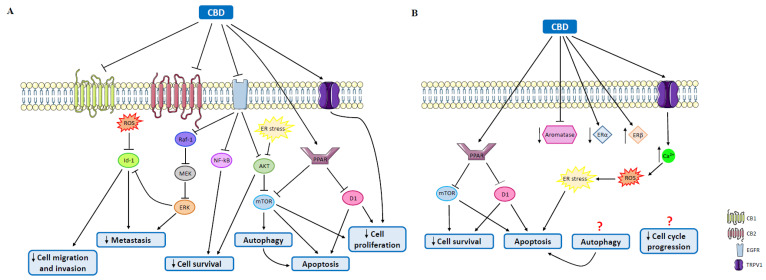
Antitumor effects mediated by CBD on TNBC and ER^+^ breast cancer. (**A**) In TNBC, CBD decreases cell proliferation by activating TRPV1 and PPAR, through the inhibition of cyclin D1 and mTOR, promoting apoptosis, which is also caused by autophagy through the induction of endoplasmic reticulum stress, leading to the inhibition of AKT and mTOR signaling. The inhibition of EGFR decreases cell survival, through the inhibition of either AKT or NF-kB, and downregulation of the Raf-1/MEK/ERK signaling pathway, which decreases the metastatic potential directly or through the inhibition of Id-1. The latter can also be promoted by ROS. (**B**) In ER^+^ breast tumors, CBD, through the activation of PPAR, induces inhibition of mTOR and cyclin D1, causing apoptosis and a reduction in cell survival. CBD can also induce apoptosis through the activation of TRPV1 via endoplasmic reticulum stress. CBD inhibits aromatase and decreases its protein levels, downregulates ERα and upregulates ERβ. D1: cyclin D1; ERα: estrogen receptor α; ERβ: estrogen receptor β; ER stress: endoplasmic reticulum stress; mTOR: mammalian target or rapamycin; PPAR: peroxisome proliferator-activated receptor; ROS: reactive oxygen species; TRPV1: transient receptor potential vanilloid 1; AKT: protein kinase B; CB1: cannabinoid receptor 1; CB2: cannabinoid receptor 2; EGFR: epidermal growth factor receptor; ERK: extracellular signal-regulated kinase; MEK: mitogen-activated protein kinase kinase; NF-kB: nuclear factor kB; Raf-1: proto-oncogene Raf-1.

**Figure 4 molecules-27-00156-f004:**
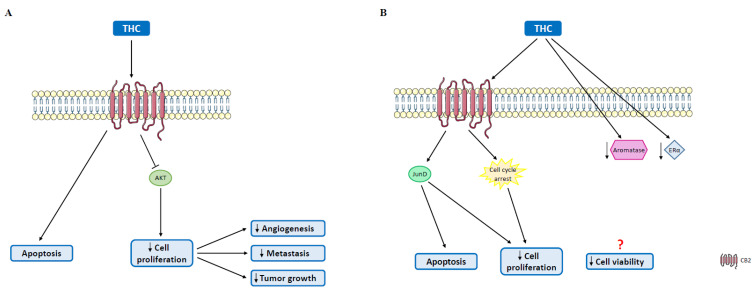
Antitumor effects mediated by THC on HER2^+^ and ER^+^ breast tumors. (**A**) In HER2^+^ tumors, THC induces apoptosis and reduces cell proliferation through AKT inhibition, which is associated with decreased angiogenic potential, and a reduction in metastasis formation and tumor growth. (**B**) In ER^+^ breast cancer cases, THC also induces apoptosis and decreases tumor cell proliferation, through JunD activation and cell cycle disruption. THC also reduces the protein levels of aromatase and ERα, decreasing cell viability. AKT: protein kinase B; CB2: cannabinoid receptor 2; ERα: estrogen receptor α; JunD: proto-oncogene JunD.

**Figure 5 molecules-27-00156-f005:**
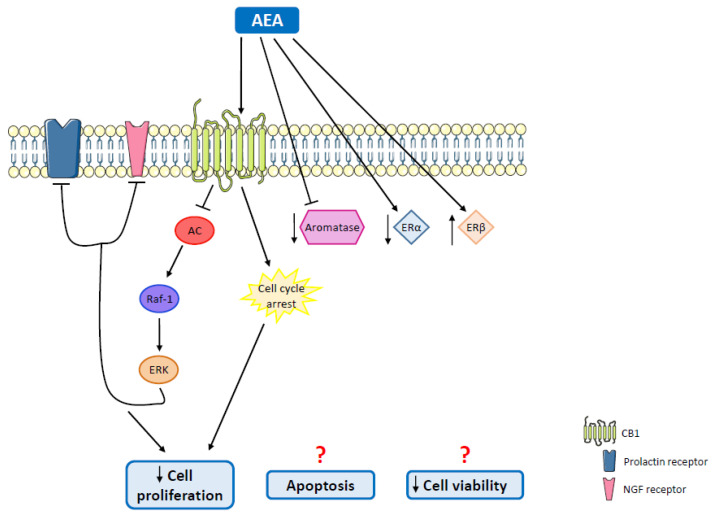
Antitumor effects mediated by AEA on ER^+^ breast tumors. The sustained activation of the Raf-1/ERK pathway, as well as cell cycle arrest, reduces cell proliferation and survival. AEA is also able to inhibit aromatase function and decrease its protein levels, to downregulate ERα and upregulate ERβ, inducing apoptosis and decreasing cell viability. AC: adenylyl cyclase; CB1: cannabinoid receptor 1; ERα: estrogen receptor α; ERβ: estrogen receptor β; ERK: extracellular signal-regulated kinase; NGF: nerve growth factor; Raf-1: proto-oncogene Raf-1.

**Table 1 molecules-27-00156-t001:** Molecular classification and prevalence of breast cancer subtypes.

Breast Cancer Subtype	HER2 Expression	ER Expression	PR Expression	Ki67 Expression	Prevalence
**Luminal A**	Negative	Positive	High	Low	73%
**Luminal B**	Positive or Negative	Positive	Low or Any	High or Any	11%
**HER2^+^**	Positive	Negative	Negative	-	12–20%
**TNBC**	Negative	Negative	Negative	-	15–20%

**Table 2 molecules-27-00156-t002:** Antitumor effects of cannabinoids in TNBC.

Model	Cannabinoid	Biological Effect	Mechanism of Action	Reference
Xenograft-Based and PyMT Genetically Engineered Models	JWH-133	Tumor growth reduction; angiogenesis inhibition	COX-2/PGE_2_ axis inhibition through CB2	[121]
MDA-MB-231 Cells	Met-F-AEA	Cell migration impairment; cell cycle arrest	Inhibition of (FAK)/Src and RhoA-ROCK pathways through CB1	[122,123]
MDA-MB-231 Cells	JWH-133	Cell migration impairment; cell cycle arrest	COX-2/PGE_2_ axis inhibition through CB2	[121]
MDA-MB-231 Cells	JWH-015	Cell migration impairment	Inhibition of ERK and cytoskeletal focal adhesion and stress fiber formation through CB2	[124]
MDA-MB-231 Cells	WIN 55,212-2	Cell migration impairment; cell cycle arrest	COX-2/PGE_2_ axis inhibition through CB2	[124]
MDA-MB-231 Cells	CBD	Cell proliferation reduction; apoptosis; autophagy	Endoplasmic reticulum stress and AKT/mTOR inhibition	[111]
MDA-MB-231 Cells	CBD	Cell proliferation reduction	TRPV1 receptors and uncharacterized CBD targets	[109,110]
MDA-MB-231 Cells	CBD	Cell proliferation reduction	Increased ROS production	[110,115,116]
MDA-MB-231 Xenografts in Immune-Deficient Mice and Orthotopic Xenografts from 4T1 Cells in Syngeneic BALB/c Mice	CBD	Cell proliferation reduction	Downregulation of Id-1	[110,115]
MDA-MB-231 and 4T1 Cells	CBD	Cell proliferation reduction	Downregulation of Id-1	[115,116]
SUM159, 4T1.2 and SPC2 Cells	CBD	Cell proliferation reduction; impairment of cell migration; invasion	Inhibition of EGF/EGFR signaling	[114]
MDA-MB-231 Cells	CBD	Apoptosis	Interplay among PPARy, mTOR and cyclin D1	[112]
MDA-MB-231 and MDA-MB436 Cells	CBG	Cell viability reduction; impairment of cell migration	Decreased Id-1 expression	[116]
MDA-MB-231 and MDA-MB436 Cells	CBN	Cell viability reduction; impairment of cell migration	Decreased Id-1 expression	[116]

**Table 3 molecules-27-00156-t003:** Antitumor effects of cannabinoids in ER^+^ breast cancer.

Model	Cannabinoid	Biological effect	Mechanism of action	Reference
MCF-7 and EFM-19 cells	AEA	Cell proliferation inhibition	Cell cycle arrest; CB1 activation; Raf-1/ERK/MAPK pathway activation	[126]
MCF-7 cells	AEA	Decreased cell proliferation	CB1 activation	[127,128]
MCF-7aro cells	AEA	Apoptosis; reduction in aromatase and ERα protein levels; upregulation of ERβ; aromatase inhibition	Cell cycle arrest	[135]
T-47D cells	CBD	Cell survival impairment; apoptosis	Interplay among PPARy, mTOR and cyclin D1	[112]
MCF-7aro cells	CBD	Apoptosis; autophagy; reduction in aromatase and ERα protein levels; upregulation of ERβ; aromatase inhibition	Cell cycle arrest	[135]
MCF-7 cells	CBD	Apoptosis	Endoplasmic reticulum stress; disruption of protein folding	[134]
EVSA-T cells	THC	Apoptosis	Cell cycle arrest mediated by CB2	[129,130]
MCF-7	THC	Decreased cell proliferation		[132]
MCF-7aro cells	THC	Apoptosis; reduction in aromatase and ERα protein levels	Cell cycle arrest	[135]
MCF-7	CBG	Decreased cell proliferation		[110]

## Data Availability

Not applicable.

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
