# Peer review of "Cannabinoids in Breast Cancer: Differential Susceptibility According to Subtype"

_molecules, 2021, doi:10.3390/molecules27010156_

Round 1

Reviewer 1 Report

This review summarized the role of cannabinoids in each breast cancer subtype and discussed the potential regulatory mechanisms. It would help us to better understand the clinical potential of cannabinoids in breast cancer subtypes. Some issues still need to be improved:

  1. In section 2. Endocannabinoid system, the authors firstly mentioned the classification, distribution and function of cannabinoid receptors (CBs), and then introduced endogenous cannabinoids. The authors are advised to introduce CBs in a separate section.
  2. In recent years, there have been some researches about synthetic cannabinoids.  Please add these contents.
  3. The novelty of this review should be further elucidated, since there are similar published publication 'Cannabinoids: a new hope for breast cancer therapy? Cancer treatment reviews, 2012, 38(7): 911-918.'

Author Response

This review summarized the role of cannabinoids in each breast cancer subtype and discussed the potential regulatory mechanisms. It would help us to better understand the clinical potential of cannabinoids in breast cancer subtypes. Some issues still need to be improved:

  1. In section 2. Endocannabinoid system, the authors firstly mentioned the classification, distribution and function of cannabinoid receptors (CBs), and then introduced endogenous cannabinoids. The authors are advised to introduce CBs in a separate section.

In the revised version of the manuscript, cannabinoid receptors (CBs) were introduced in a separate section. Thus, we have now introduced CBs in section 2 (page 3) and the information related to endocannabinoid system is now in section 3 (page 3).

  1. In recent years, there have been some researches about synthetic cannabinoids.  Please add these contents.

We thank the reviewer for this suggestion. We have introduced, in the revised version of the manuscript, in section 1, pages 2-3, general information about synthetic cannabinoids. It is important to note that detailed information about the effects of synthetic compounds on breast cancer are referred on section 5.1, pages 9-10, in which it is addressed the knowledge obtained so far on the effects of synthetic cannabinoids on triple-negative breast cancer.

  1. The novelty of this review should be further elucidated, since there are similar published publication 'Cannabinoids: a new hope for breast cancer therapy? Cancer treatment reviews, 2012, 38(7): 911-918.'

We agree with the reviewer, since the referred publication is, in fact, in the research area of the one we are presenting. However, it is important to take into account that “Cannabinoids: a new hope for breast cancer therapy?” was published in 2012, almost 10 years ago. During these years, several studies have been conducted and more knowledge related to the effects of cannabinoids on the different breast cancer subtypes, especially regarding estrogen receptor-positive breast cancer, has been obtained. Thus, it is important to summarize all the novelties discovered so far in order to help other researchers interested in cannabinoids and breast cancer.

Reviewer 2 Report

Authors have demonstrated the role of Cannabinoids in breast cancer. I have just a few suggestions

1.This mauscript needs linguistic improvement.

2.Some citations and reference are missing

In Page 8:"Moreover, it has been demonstrated, in MCD-MB-231 cells, that CBD induces the production of reactive oxygen species (ROS) [107, 108, 112, 113], which are able to downregulate
Id-1 activity (Figure 3A), corroborating the apoptotic effects previously mentioned." ROS plays a very important role in cancer development, neurologic disorders and so on, especially for breast cancer, please add more information about it.  (Please cite: 1. Chen et al. Semin Cancer Biol. 2020 Oct 6:S1044-579X(20)30203-0. doi: 10.1016/j.semcancer.2020.09.012.  2. Shekhar et al. International Journal of Molecular Sciences. 2021; 22(4):2074. https://doi.org/10.3390/ijms22042074)

Author Response

Authors have demonstrated the role of Cannabinoids in breast cancer. I have just a few suggestions:

  1. This mauscript needs linguistic improvement.

We thank the reviewer for this advertisement. We made a linguistic revision in the new version of the manuscript.

  1. Some citations and reference are missing In Page 8:"Moreover, it has been demonstrated, in MCD-MB-231 cells, that CBD induces the production of reactive oxygen species (ROS) [107, 108, 112, 113], which are able to downregulate Id-1 activity (Figure 3A), corroborating the apoptotic effects previously mentioned." ROS plays a very important role in cancer development, neurologic disorders and so on, especially for breast cancer, please add more information about it.  (Please cite: 1. Chen et al. Semin Cancer Biol. 2020 Oct 6:S1044-579X(20)30203-0. doi: 10.1016/j.semcancer.2020.09.012.  2. Shekhar et al. International Journal of Molecular Sciences. 2021; 22(4):2074. https://doi.org/10.3390/ijms22042074)

We thank the reviewer for this suggestion. In the revised version of the manuscript, the role of ROS in breast cancer development was addressed in section 5.1, page 8, and Chen et al., Semin Cancer Biol, 2020 was cited. We did not insert Shekhar et al., International Journal of Molecular Sciences, 2021, because this paper is not related with the subject of the work.

Round 2

Reviewer 1 Report

In the revised of the manuscript, the authors have been added some contents about synthetic cannabinoids, but these contents were not added in an appropriate position. The authors are advised to introduce these contents in section 1, passage 2.

Author Response

In the revised of the manuscript, the authors have been added some contents about synthetic cannabinoids, but these contents were not added in an appropriate position. The authors are advised to introduce these contents in section 1, passage 2.

As suggested, in the new version of the manuscript, the contents related to synthetic cannabinoids are in the midle of section 1, page 2.

Reviewer 2 Report

Strongly suggest to publish.

Author Response

Strongly suggest to publish.

Thank you!